# Melatonin Alleviates Lipopolysaccharide-Induced Abnormal Pregnancy through MTNR1B Regulation of m6A

**DOI:** 10.3390/ijms25020733

**Published:** 2024-01-05

**Authors:** Shisu Zhao, Yanjun Dong, Yuanyuan Li, Zixu Wang, Yaoxing Chen, Yulan Dong

**Affiliations:** 1Laboratory of Neurobiology, College of Veterinary Medicine, China Agricultural University, Beijing 100193, China; zhaoshisuzss@163.com (S.Z.); yanjund@cau.edu.cn (Y.D.); lyydyxzz@163.com (Y.L.); zxwang2007@163.com (Z.W.); 2Key Laboratory of Precision Nutrition and Food Quality, Ministry of Education, China Agricultural University, Beijing 100193, China

**Keywords:** melatonin, m6A, pregnancy, uterus, inflammation

## Abstract

Pregnancy is a highly intricate and delicate process, where inflammation during early stages may lead to pregnancy loss or defective implantation. Melatonin, primarily produced by the pineal gland, exerts several pharmacological effects. N6-methyladenosine (m6A) is the most prevalent mRNA modification in eukaryotes. This study aimed to investigate the association between melatonin and m6A during pregnancy and elucidate the underlying protective mechanism of melatonin. Melatonin was found to alleviate lipopolysaccharide (LPS)-induced reductions in the number of implantation sites. Additionally, it mitigated the activation of inflammation, autophagy, and apoptosis pathways, thereby protecting the pregnancy process in mice. The study also revealed that melatonin regulates uterine m6A methylation levels and counteracts abnormal changes in m6A modification of various genes following LPS stimulation. Furthermore, melatonin was shown to regulate m6A methylation through melatonin receptor 1B (MTNR1B) and subsequently modulate inflammation, autophagy, and apoptosis through m6A. In conclusion, our study demonstrates that melatonin protects pregnancy by influencing inflammation, autophagy, and apoptosis pathways in an m6A-dependent manner via MTNR1B. These findings provide valuable insights into the mechanisms underlying melatonin’s protective effects during pregnancy and may have implications for potential therapeutic strategies in managing pregnancy-related complications.

## 1. Introduction

Compared to animals, humans have a lower success rate for pregnancy of approximately 30%, with 75% of pregnancy failures attributed to implantation failures [1]. Embryo implantation plays a crucial role in the reproductive process. Various factors can contribute to implantation failure, such as maternal endocrine disorders, abnormal uterine development, inflammation, and stress [2]. Apart from the immune system, several physiological systems are involved in orchestrating pregnancy, including metabolic, endocrine, and circadian systems [3].

Melatonin, a neurohormone, exhibits a wide range of regulatory and protective effects, including the synchronization of circadian rhythms [4], modulating of the immune system [5], and regulating of reproductive function [6]. Apart from its role in regulating circadian rhythms [7], researchers have demonstrated that melatonin is present in high levels in human preovulatory follicular fluid [8] and plays a role in the timing of labor [9]. Studies have revealed that melatonin enhances embryo implantation by increasing the E2 levels during pregnancy [10] and that melatonin receptor MT2 influences early gestation [11]. However, the precise mechanisms underlying the beneficial effects of melatonin on reproduction remain to be further elucidated.

N6-methyladenosine (m6A) is the most prevalent internal modification found in mammalian RNA molecules [12]. The regulators of m6A include methyltransferases, demethylases, and binding proteins, commonly referred to as writers, erasers, and readers, respectively. Notably, writers consist of METTL3/14/16, WTAP, and VERMA [13], while erasers comprise FTO and ALKBH5 [14]. As for the readers, YTHDF1/2/3, YTHDC1/2, and IGF2BP1/2/3 are the ones that selectively recognize m6A on mRNA [15]. Increasing evidence suggests that m6A methylation plays essential roles in various gynecological diseases, such as adenomyosis [16] and endometriosis [17]. In our previous studies, we observed that the m6A levels in the uterus increased as pregnancy progressed [18], indicating that m6A methylation may play a crucial role in the establishment of implantation and maintenance of pregnancy. 

Although some studies have shown that melatonin affects the female reproductive system, it remains unclear whether the protective effect of melatonin is exerted through m6A methylation. In this study, we aimed to investigate the protective effects of melatonin on LPS-induced abnormalities during early pregnancy. Our data demonstrate that melatonin protected mice from LPS-induced abnormal pregnancy and alleviated inflammation, autophagy, and apoptosis via MTNR1B-m6A pathway.

## 2. Results

### 2.1. Melatonin Protects against LPS-Induced Early Pregnancy Abnormalities in Mice

Mice were treated as described in Figure 1A. Daily water consumption, feed intake, and body weight of mice were monitored from D1 to D5 of gestation. The levels of daily water consumption (*p* < 0.0001), feed intake (*p* < 0.0001), and weight (*p* < 0.0001) of mice decreased immediately after LPS injection, and melatonin alleviated this decrease (Figure 1B). Furthermore, a significant decrease in the number of implantation sites in the uterus was observed in the LPS group (*p* < 0.0001), while the number of implantation sites and the weight of the uterus in the Mel+LPS group were higher than those in the LPS group (*p* = 0.0109) (Figure 1C–E). Additionally, melatonin was found to attenuate the LPS-induced decrease in blood glucose values (Figure 1F), as well as the reduction in platelet and white blood cell counts (Figure 1G). Moreover, melatonin was effective in alleviating the abnormal secretion of serum hormones induced by LPS, including estrogen (E2), progesterone (P4), corticosterone (CORT), and norepinephrine (NOR) (Figure 1H).

### 2.2. Melatonin Alleviates LPS-Induced Inflammation, Autophagy and Apoptosis in the Uterus

To investigate the effect of LPS on gene expression in the uterus, transcriptome sequencing data from LPS-challenged and control mice were obtained from the GEO database. We identified 146 differentially expressed genes (DEGs) between the two groups (Appendix A). GO and KEGG analyses revealed that these DEGs were mainly enriched in immunity and inflammation-related biological processes and pathways (Figure 2A,B). Furthermore, we examined the expression of 23 cytokines in the uterus using a cytokine assay method. The results demonstrated that melatonin significantly reduced the elevation of cytokines induced by LPS stimulation (Figure 2C and Appendix A). Similar patterns were observed in the serum as well (Appendix A). Subsequently, we analyzed inflammation-, autophagy-, and apoptosis-related genes in the decidua of LPS-challenged mice from the GEO database and found that these genes were upregulated in the LPS-treated group (Figure 2D–F). In this study, we also detected the mRNA levels of several genes associated with inflammation, autophagy, and apoptosis processes (Figure 2G–I). The results showed that inflammation-related genes *Tlr4* (*p* = 0.0432), *Myd88* (*p* = 0.0220), and *Rela* (*p* = 0.0052) were significantly increased in the LPS group compared to the Veh group, while Mel+LPS had no significant effect compared to the Veh group. Additionally, melatonin alleviated the abnormal expression of autophagy and apoptosis genes after LPS stimulation, such as *Sqstm1*, *Becn1*, *Atg5*, *Atg7*, *Bcl2*, *Bax*, *Casp3* and *Casp9*.

### 2.3. Melatonin Alleviates Elevated m6A Levels Induced by LPS in the Uterus

To investigate whether LPS affects m6A modification, we analyzed the expression levels of m6A regulator genes in the decidua of LPS-challenged mice from the GEO database. The analysis showed that the m6A writers *Mettl16* (*p* = 0.04589) and the readers *Ythdc1* (*p* = 0.0963), *Ythdf2* (*p* = 0.0006), and *Ythdf3* (*p* < 0.0001) were upregulated in the LPS group (Figure 3A). In our study, we detected the m6A levels in the implantation sites and found that melatonin alleviated the increase in m6A levels caused by LPS (Figure 3B), as well as the mRNA levels of m6A writers *Mettl3* and *Mettl14* (Figure 3C). Furthermore, we observed stronger expression of METTL3 in the LPS group (Figure 3D). Moreover, we analyzed the protein–protein interactions through STRING analysis, which suggested that melatonin receptor 1B (MTNR1B) was an interacting protein of the m6A recognition proteins (Figure 3E). The mRNA levels of *Mtnr1b* in the uterus were detected, and the results indicated that the expression of Mtnr1b tended to be enhanced after melatonin injection, suggesting that melatonin may function through its receptors (Figure 3F). The localization of METTL3, FTO, and MTNR1B revealed their specific expression in endometrial stromal cells (Figure 3G). Next, we performed m6A-seq to detect RNA from the uterus in the Veh, LPS, and Mel+LPS groups. We found that the differential m6A peaks were mainly enriched in the 3′UTR (Figure 4A,B). Additionally, we analyzed the m6A motifs that changed after melatonin and LPS treatment (Figure 4C). Subsequently, we identified differentially methylated genes between groups (Figure 4D). Further, we performed KEGG pathway analysis on the m6A-up genes in the LPS group, which showed pathways enriched in metabolic pathways and the NF-κB signaling pathway (Figure 4E). Additionally, GO enrichment for m6A-up genes in the LPS group was conducted (Figure 4F). Subsequently, we analyzed the intersection of m6A-up genes in the LPS group and m6A-down genes in the Mel+LPS group (Figure 4G). We also analyzed the intersection of the m6A-down genes in the LPS group and the m6A-up genes in the Mel+LPS group (Figure 4H). The results indicated that m6A modification of many genes was altered after melatonin treatment.

### 2.4. Melatonin Alleviates LPS-Induced Inflammation, Autophagy, and Apoptosis in HESCs

In the next step, we conducted in vitro experiments using human endometrium cells (HESCs) to study the mechanism of melatonin. We performed a CCK8 assay to determine the effective concentrations of melatonin and LPS. For subsequent experiments, we used melatonin at 1 μM and LPS at 50 μg/mL for 48 h (Figure 5A). Subsequently, we assessed the protein levels of several genes associated with inflammation, autophagy, and apoptosis processes. In particular, inflammation-related proteins p-RELA (*p* = 0.0007) and ERK1/2 (*p* = 0.0090), autophagy-related proteins LC3B (*p* = 0.0377) and ATG7 (*p* = 0.0001), and apoptosis-related proteins c-PARP (*p* = 0.0085), BAX (*p* = 0.0080), and CASP1 (*p* = 0.0188) were significantly increased in the LPS group compared to the Veh group. However, co-treatment with Melatonin and LPS (Mel+LPS) showed no significant effect compared to the Veh group, indicating that melatonin counteracted the LPS-induced activation of these pathways (Figure 5B–D). Similarly, the results of the transcript level test also revealed that melatonin inhibited the activation of inflammation, autophagy, and apoptosis pathways induced by LPS (Appendix A–C). Overall, the in vitro experiments yielded consistent findings with the outcomes observed in the in vivo tests.

### 2.5. Melatonin Alleviates LPS-Induced Elevated m6A Levels in HESCs

To explore the effects of LPS and melatonin on the m6A level of HESCs, we extracted total RNA and detected m6A levels using LC-MS/MS (Figure 6A). The results revealed that melatonin alleviated the increase in m6A levels induced by LPS. Additionally, the mRNA levels of m6A regulators, including *METTL3*, *METTL14*, *ALKBH5*, and *YTHDF1*, were also influenced by melatonin, showing a reduction compared to the LPS group (Figure 6B). Subsequently, we examined the protein levels of m6A regulators and observed that METTL3 was expressed more strongly in the LPS group (Appendix A). Next, we conducted immunofluorescence to localize the proteins METTL3 and FTO in HESCs. Interestingly, METTL3 was strongly expressed in the nucleus of HESCs, regardless of the treatment conditions. However, the localization of FTO in the nucleus was reduced after LPS treatment, suggesting that melatonin may regulate m6A modification by controlling the entry of FTO into the nucleus (Figure 6C). We then analyzed the ratio of fluorescence intensity in the nucleus and found that the localization of FTO in the nucleus decreased after LPS stimulation, while it increased after melatonin addition, possibly due to its demethylation function (Figure 6E,F). Furthermore, we localized the protein MTNR1B in HESCs and found that MTNR1B was present in the nucleus, cytoplasm, and cell membrane of HESCs (Figure 6D). Interestingly, LPS and melatonin did not affect the nuclear expression of MTNR1B (Figure 6G). Additionally, we examined the transcriptional level of MTNR1B in HESCs and found that the addition of melatonin activated the expression of MTNR1B (Figure 6H). These findings suggest that melatonin exerts its regulatory effects on m6A modification in HESCs by influencing the expression and subcellular localization of m6A regulators, as well as through the activation of MTNR1B. These insights provide valuable information regarding the mechanisms underlying the protective effects of melatonin on m6A dynamics in response to LPS-induced changes.

### 2.6. Melatonin Plays a Protective Role through MTNR1B

To investigate whether melatonin exerts its protective effects through its receptor and the m6A pathway, we used the melatonin receptor antagonist 4-P-PDOT and the METTL3-METTL14 complex inhibitor SAH. Cell viability was assessed, and subsequent treatment concentrations of 0.1 μM of 4-P-PDOT and 1 μM of SAH were utilized (Figure 7A). Next, we detected the m6A levels using LC-MS/MS, and the results demonstrated that the ability of melatonin to regulate the level of m6A was weakened after antagonizing MTNR1B (Figure 7B). Subsequently, we observed that inflammation-, autophagy-, and apoptosis-related genes were significantly increased in the LPS group compared to the vehicle group, while the Mel+LPS group showed no significant effect compared to the Veh group (Figure 7C–H). Additionally, these trends were confirmed by flow cytometric analysis (Figure 7I). These findings suggest that melatonin plays a protective role in HESCs through the MTNR1B receptor and the m6A pathway. When MTNR1B is antagonized, the ability of melatonin to regulate m6A levels and protect against LPS-induced alterations in inflammation, autophagy, and apoptosis is diminished.

## 3. Discussion

Embryo implantation is a critical step in establishing a successful pregnancy, and failed implantation is a significant obstacle in assisted reproduction [1]. There is a substantial body of evidence supporting the beneficial effects of melatonin on pregnancy [19]. Previous studies have indicated that melatonin can enhance the implantation ability of mouse embryos, possibly through the binding of uterine HB-EGF to ErbB1 and ErbB4 receptors [20]. Another experiment demonstrated that melatonin improves embryo implantation and increases litter size by elevating E2 levels [10]. While the positive effects of melatonin on pregnancy and embryo implantation are well-established, the exact mechanisms underlying its impact on inflammation in pregnancy require further investigation. Further research in this area may provide valuable insights into the potential therapeutic applications of melatonin in improving reproductive outcomes and addressing implantation-related challenges.

In this study, LPS was utilized to induce abnormal pregnancy in mice during early pregnancy, and the protective effect of melatonin on pregnancy was investigated. The implantation of mouse embryos was observed at D5, and it was found that melatonin protected against the reduction in implantation sites induced by LPS stimulation. Previous reports have indicated that the addition of exogenous melatonin can down-regulate rat E2 levels, up-regulate P4 levels, promote oocyte maturation in vitro, and facilitate the implantation of transplanted embryos [21]. In this study, melatonin was found to alleviate the decrease in E2 and P4 levels and the increase in CORT and NOR levels induced by LPS. Earlier studies have demonstrated that LPS can induce an inflammatory response and activate the NF-κB signaling pathway [22]. It has been reported that melatonin’s anti-inflammatory mechanism may involve inhibiting TLR4 signaling [23]. Similarly, in this study, melatonin was found to alleviate the activation of the NF-κB signaling pathway caused by LPS. In other investigations, the regulatory role of melatonin in autophagy has been observed in disease models [24]. In line with these findings, our study examined autophagy pathway-related genes and found that melatonin decreased the expression of Sqstm1, Becn1, and Atg7 after LPS stimulation. This suggests that melatonin actively regulates autophagy. There is a substantial body of literature demonstrating the interaction between autophagy and apoptosis, with some studies showing that autophagy inhibits apoptosis [25]. Melatonin has also been reported to prevent apoptosis by reducing CASP3 activation and DNA damage [26]. In this study, apoptosis pathway-related genes were examined in the uterus of melatonin and LPS-treated mice. The results showed that melatonin could alleviate the abnormal increase of apoptosis pathway-related genes *Bax* and *Casp9* under LPS stimulation.

With the advancement of epigenetics research, it has been revealed that epigenetic regulations, such as DNA methylation [27] and histone modification [28], play crucial roles in the female reproductive system. Now, RNA modification has emerged as a new direction for studying reproductive functions. Studies have demonstrated that m6A, a type of RNA modification, is also important in reproductive processes. For instance, one study found that the level of m6A in the endometrium of patients with adenomyosis was significantly reduced [16]. Dysregulation of m6A regulators has also been reported in endometriosis [17]. Additionally, recent research has shown that melatonin can modulate the pluripotency of stem cells through the m6A pathway [29]. However, the mechanisms by which m6A impacts the pregnancy process have been scarcely explored, and whether melatonin exerts its effects on improving pregnancy through the m6A pathway remains largely unknown. 

In this study, a protein interaction network analysis revealed that the melatonin receptor MTNR1B interacts with m6A-related proteins, suggesting that melatonin may influence m6A modification through MTNR1B. The m6A levels in the uterus were then measured, and it was observed that m6A levels increased after LPS stimulation, while melatonin treatment alleviated this abnormal increase. Subsequently, the transcription levels of m6A-related genes were examined, and it was found that melatonin mitigated the up-regulation of methylated genes *Mettl3* and *Mettl14*. The results of protein detection also indicated an increase in the expression level of METTL3. Interestingly, another study also observed that LPS stimulation led to an up-regulation of m6A and METTL3 levels in human dental pulp cells [30]. Taken together, these findings suggest that melatonin alleviates the LPS-induced increase in m6A levels in the uterus of mice by regulating m6A methylation-related genes. The in vivo results demonstrated that intraperitoneal injection of melatonin in mice could alleviate LPS-induced abnormal pregnancy and inhibit the abnormal increase of m6A modification, inflammation, autophagy, and apoptosis in the uterus of mice.

Based on the research results mentioned above, this study further explored the specific mechanism of melatonin using human endometrial stromal cells in vitro. Recent findings have indicated that melatonin regulates stem cell pluripotency through the MT1-JAK2/STAT3-Zfp217 signaling axis [29]. In this study, the results suggested that melatonin may affect m6A modification through its membrane receptor MTNR1B. To investigate this further, the researchers used an MTNR1B antagonist, 4-P-PDOT, and a METTL3-METTL14 complex inhibitor, SAH, for cell treatments. Interestingly, the stabilizing effect of melatonin on m6A levels was inhibited when 4-P-PDOT was added, indicating that melatonin’s impact on m6A may be mediated through its receptor MTNR1B. In this study, after treating cells with melatonin and LPS, it was found that melatonin could alleviate the LPS-induced increases in mRNA levels of inflammation-related genes *TLR4*, *MYD88*, *RELA*, *STAT3*, as well as the protein levels of RELA, p-RELA, and ERK1/2. A similar observation was made in a colitis study in rats, where melatonin reduced the up-regulation of TLR4, MyD88, and NF-κB induced by trinitrobenzenesulfonic acid [31]. Interestingly, in the present study, when MTNR1B was antagonized, the alleviating effect of melatonin on LPS-induced inflammation was absent. This suggests that melatonin’s ability to attenuate inflammation in response to LPS may be dependent on its interaction with the MTNR1B receptor.

This study also conducted a series of tests on the autophagy pathway and found that melatonin could alleviate the LPS-induced increase in mRNA levels of autophagy-related genes *SQSTM1*, *LC3B*, *BECN1*, *ATG5*, *ATG7*, as well as the protein levels of LC3B, ATG5, and ATG7. However, when MTNR1B was antagonized, the alleviating effect of melatonin disappeared. Similarly, apoptosis-related genes *BAX*, *BCL2*, *BIM*, *CASP3*, and *CASP9* also showed increased mRNA levels after LPS stimulation, which were reduced by melatonin treatment. The protein levels of c-PARP, BCL2, CASP1, and c-CASP3 followed a similar pattern. Notably, the protective effect of melatonin on apoptosis was also inhibited after the addition of an MTNR1B antagonist, as confirmed by flow cytometry. Furthermore, the study detected the effects of SAH, the METTL3-METTL14 complex inhibitor, on inflammation-, autophagy-, and apoptosis-related proteins. Interestingly, the expression of these proteins was significantly reduced after the addition of SAH. This could be attributed to the fact that m6A is a crucial post-transcriptional modification that regulates mRNA translation [15]. When m6A methylation is inhibited, these genes may not be translated into proteins normally. Additionally, the study found that cell apoptosis was significantly increased after the addition of SAH, which aligns with previous reports indicating that inhibition of m6A modification can lead to increased apoptosis. In zebrafish embryos, for example, METTL3 knockout resulted in tissue differentiation defects and increased apoptosis [32]. Studies have also reported that silencing METTL3 in human osteosarcoma cells significantly inhibits cell proliferation, migration, and invasion and promotes apoptosis [33]. Similarly, there are reports of increased apoptosis following inhibition of the catalytic activity of METTL3-METTL14 [34]. These findings collectively suggest that under normal circumstances, appropriate m6A modification maintains the physiological function of cells, while an excessive or insufficient level of m6A modification may not be conducive to cell survival.

This murine study highlights melatonin’s potential in reproductive health; however, translating findings to humans demands careful validation. Melatonin’s interaction with vital pathways like inflammation, autophagy, apoptosis, and m6A warrants clinical exploration. Targeted therapies leveraging MTNR1B modulation offer promise. While promising, human-centric investigations are essential to confirm melatonin’s clinical utility in reproductive medicine.

## 4. Materials and Methods

### 4.1. Animals and Organ Collection

Female ICR mice of 8 weeks old were purchased from Vital River Laboratory Animal Technology (Beijing, China). All animal procedures were approved by the Animal Care and Use Committee of China Agricultural University (ethics approval number: AW03602202-2-1). Mice were bred under controlled environmental conditions (12 h light/dark cycle, relative humidity 55–65% and 23–27 °C). Female mice were mated to males at a ratio of 2:1. The day of mating was counted as day 0 (D0), and a vaginal plug was used to mark the first day of pregnancy (D1). As shown in Figure 1A, the pregnant mice were randomly divided into four groups: injected with vehicle (Veh), injected with LPS (LPS), injected with melatonin and LPS (Mel+LPS), injected with melatonin (Mel). The LPS (Sigma-Aldrich, St. Louis, MI, USA) injection volume was 0.5 mg/kg body weight dissolved in 0.9% sterile saline. The melatonin (Sigma-Aldrich, St. Louis, MI, USA) injection volume was 10mg/kg body weight dissolved in 0.9% sterile saline containing 3% ethanol. On D1 to D4, mice were received intraperitoneal injections of melatonin or vehicle daily at 9:00 a.m., and on days 3 and 4, LPS was injected intraperitoneally 30 min after the melatonin injection. The mice were sacrificed on D5, and the uterus was harvested. Blood samples were harvested, and blood routine examination was performed using the whole blood by a routine blood test instrument (Mindray, Shenzhen, China). Implantation sites were visualized after tail vein injection of 1% Chicago blue (Sigma-Aldrich, St. Louis, MI, USA).

### 4.2. Cell Culture and Treatment

Immortalized human endometrial stromal cell line (HESCs), CRL-4003, was a kind gift from Prof. Renwei Su (South China Agricultural University, Guangzhou, China). Cells were cultured in DMEMB/F12 (Gibco, Grand Island, NE, USA) containing 10% FBS (Gibco, Grand Island, NE, USA). Once cells were attached, HESCs were starved for 12 h in DMEM/F12, cells were treated with 1 μM Melatonin and 50 μg/mL LPS for 48 h. For experiments involving inhibitors, cells were incubated with inhibitors for 48 h. The following specific inhibitors were used: melatonin receptor antagonist 4-P-PDOT (MedChemExpress, Monmouth Junction, NJ, USA), and the METTL3-METTL14 complex inhibitor SAH (MCE, Monmouth Junction, NJ, USA). 4-P-PDOT was used at a concentration of 0.1 μM and SAH was used at a concentration of 1 μM.

### 4.3. Blood Index Detection Method

Routine blood examination indices were detected via a blood cell analyser (NIHON KOHDEN, Tokyo, Japan) in whole blood. The blood glucose of the tail vein was detected by a Yuyue blood glucose meter (Yuyue, Shanghai, China). The levels of progesterone (P4), Estradiol-17β (E2), corticosterone (CORT), and noradrenaline (NOR) were measured spectrophotometrically in serum using radioimmunoassay kit according to the manufacturer’s instructions. These kits were purchased from Beijing Huaying Institute of Biotechnology (Huaying, Beijing, China).

### 4.4. Luminex Liquid Suspension Chip Detection

The procedure was performed according to the instructions of Bio-Plex Pro Mouse Cytokine 23-plex (Bio-Rad, Hercules, CA, USA). Serum was diluted 4 times with Sample Diluent, and tissue lysate samples were diluted 25 times with Sample Diluent. Finally, 50 μL of the diluted samples was loaded for detection, and then the standard was diluted according to the instructions. We diluted the microbeads with assay buffer. After the diluted microbeads are shaken, we added 50 μL of each well to a 96-well plate, washed three times with a plate washer, and added 50 μL of the prepared standards, samples, and blanks to the 96-well plate, placed them on a plate shaker protected from light, and incubated at room temperature for 30 min. We discarded the sample, washed the plate 3 times, added 25 μL of diluted detection antibody to each well, placed this on a plate shaker to avoid light, and incubated at room temperature for 30 min. We discarded the detection antibody, washed 3 times, used assay buffer to dilute Streptavidin-PE according to the instructions, added 50 μL of diluted Streptavidin-PE to each well, placed on a plate shaker to avoid light, incubated at room temperature for 10 min, and each well washed 3 times. The wells were resuspended by adding 125 μL assay buffer, placed on a plate shaker at room temperature, shaken for 30 s in the dark, and detected by Bio-Plex MAGPIX System (Bio-Rad, Hercules, CA, USA).

### 4.5. Quantitative Analysis of m6A Level Using LC-MS/MS

Total m6A was measured in 1 μg of total RNA extracted from the uterus using liquid chromatography-tandem mass spectrometry (LC-MS/MS) (Shimadzu Corporation, Kyoto, Japan). RNA was heated at 95 °C for 5 min, followed by 2 min of cooling on ice. Subsequently incubated with 1 μL of S1 nuclease (Takara, Okasa, Japan) for 4 h at 37 °C. Then, 1 μL of alkaline phosphatase (Takara, Okasa, Japan) was added, and the reaction was incubated for 1 h at 37 °C. The reaction mixture was extracted with chloroform, freeze-dried at −40 °C for 24 h, and then dissolved in 100 μL of ultra-pure water post-centrifugation. HPLC separation was performed using a C18 column (Shimadzu Corporation, Kyoto, Japan) with a flow rate of 0.2 mL/min at 35 °C. Solvent A was 0.1% (*vol*/*vol*) formic acid in water, and solvent B was 0.1% (*v*/*v*) formic acid in methanol. A gradient of 5 min of 5% B, 10 min of 5–30% B, 5 min of 30–50% B, 3 min of 50–5% B, and 17 min of 5% B was used.

### 4.6. m6A-Seq

m6A-seq was performed as previously described [12]. Briefly, 100 μg total RNA was extracted from the uterus using TRIzon Reagent (CWBIO, Taizhou, Jiangsu, China). mRNAs were fragmented into about 100-nt fragments and immunoprecipitated (IP) with 5 μg m6A antibody (Abcam, Cambridge, UK). The antibody-RNA complex was isolated by incubation with protein A beads (Invitrogen, Carlsbad, CA, USA). All libraries were sequenced on Illumina Hiseq X10 (Illumina, San Diego, CA, USA). Raw reads from Illumina sequencing were subjected to adaptor trimming and filtering of low-quality reads by Fastp (version 0.12.3). STAR (version 2.5.2a) was used to output a sorted genome-coordinate based Bam file. The exomePeak2 (version 4.2) was used to perform peak calling analysis on the IP and Input file. Difference peaks were analyzed using MACS2 (version 3.4). Peaks were annotated using CHIPseeker (version 1.22.1). Motif analysis was performed using HOMER (version 4.10). All sequencing data were uploaded to Gene Expression Omnibus (GEO series record GSE216994).

### 4.7. Identification and Functional Assessment of DEGs

Tissue-specific mRNA expression data were obtained from publicly available datasets (GSE152343). Differential expression analysis between the LPS and control groups was conducted using the EdgeR package (version 3.14.0) and limma package (version 3.30.7), with criteria set at a *p*-value < 0.05 and fold change > 2 to identify differentially expressed genes (DEGs). Heatmap visualization was generated using the pheatmap package (version 0.7.7). For functional analysis, Gene Ontology (GO) enrichment and Kyoto Encyclopedia of Genes and Genomes (KEGG) pathway analyses were performed utilizing the “clusterProfiler”, “enrichplot”, and “org.Hs.eg.db” packages, employing thresholds of *p*-value < 0.05 and *q*-value < 0.05 for significance.

### 4.8. Protein–Protein Interaction (PPI) Network Construction

The PPI network analysis of the identified DEGs was established using the STRING online database (http://string-db.org) (accessed on 20 November 2022). The detection of functional modules within the interaction networks was conducted employing the Markov clustering algorithm. Stringent screening criteria were applied to ensure high confidence (confidence > 0.9) in the selection of protein interactions.

### 4.9. Immunohistochemistry

Paraffin-embedded tissue sections were used for examination of hematoxylin-eosin (HE) staining. For IHC staining, tissue sections were repaired in 0.1 M sodium citrate buffer, then washed with PBS, and incubated in 3% hydrogen peroxide at room temperature for 25 min in darkness. Following blocking with 5% horse serum at 37 °C for 30 min, they then incubated with the following primary antibodies: anti-METTL3 (1:100, Proteintech, Wuhan, China), anti-FTO (1:100, Proteintech, Wuhan, China), anti-MTNR1B (1:100, Cell Signaling Technology, Danvers, MA, USA). Then, the slides were incubated with the secondary antibody. Chromogen detection was carried out with the DAB chromogen kit (Zhongshan Golden Bridge Biotechnology, Beijing, China). All images were taken using a microscope (Olympus, Tokyo, Japan).

### 4.10. Immunofluorescence

Immunofluorescence staining was performed at room temperature. Cells were washed in PBS, then fixed with 4% paraformaldehyde for 30 min at room temperature and washed 3 times in PBS. Cells were permeabilized with 0.1% Triton-X100 (Sigma T8787) and washed 3 times in PBS. Then, they were blocked in 5% goat serum in 37 °C for 30 min. Cells were blocked with 5% goat serum and then incubated with the following primary antibodies: anti-METTL3 (1:100, Proteintech, Wuhan, China), anti-FTO (1:100, Proteintech, Wuhan, China), anti-MTNR1B (1:100, abcam, Cambridge, UK). Then, cells were incubated with secondary antibody: goat anti-rabbit IgG H&L Alexa Fluor 488 (1:1000, cell signaling technology, Danvers, MA, USA), for 1 h in the dark. Coverslips were mounted to slides with DAPI. All images were taken using a microscope (Olympus, Tokyo, Japan).

### 4.11. Total RNA Extraction and Quantitative Real-Time PCR (qPCR)

Total RNA was isolated with TRIzon Reagent (CWBIO, Jiangsu, China). Then, cDNA was synthesized with HiScript QRTsupermix for qPCR (+gDNA wiper) (Vazyme, Nanjing, China). Real-time quantitative PCR (qPCR) was performed with SYBR green master mix (Vazyme, Nanjing, China). Changes in fluorescence were monitored on a OneStep Plus instrument (Applied Biosystems, Waltham, MA, USA), amplification conditions were as follows: initial denaturation at 95 °C for 5 s; 40 cycles of denaturation at 95 °C for 10 s, annealing at 60 °C for 30 s; melting curve analysis at 95 °C for 15 s, 60 °C for 60 s, and 95 °C for 15 s. Relative gene expression was obtained by normalizing the expression results to *Gapdh* expression (the list of primers used are shown in Appendix A).

### 4.12. Western Blot

Total protein was obtained from cells and tissues in RIPA lysis buffer (Biyuntian, Shanghai, China) with protease inhibitor (CWBIO, Taizhou, Jiangsu, China) and then separated by 10% SDS-PAGE. Membranes were blocked with 5% milk and then incubated with the following primary antibodies (for all antibodies, see Appendix A), the concentrations were all 1:1000. Then, the membranes were incubated with the secondary antibody anti-mouse or anti-rabbit horseradish peroxidase (HRP)-conjugated secondary antibodies (CWBIO, Jiangsu, China), the concentrations were 1:8000. Imaged with a Sapphire Biomolecular Imager (Azure Biosystems, Dublin, OH, USA).

### 4.13. CCK-8 Cell Viability Assay

Cell counting kit 8 (CCK8) experiment for cell viability assay were performed using the CCK8 Kit (TargetMol, Boston, MA, USA). A 100 μL cell suspension was inoculated into a 96-well plate. The cells to be tested were removed, and 10 μL of the assay reagent was directly added to the cell culture medium. The cells were further incubated for 3 h. Before spectrophotometric readings were taken, the plate was shaken on a shaker for 1 min to ensure uniform color distribution. Absorbance was measured at 450 nm.

### 4.14. Flow Cytometry to Detect Cell Apoptosis

To quantify the cell apoptotic rate, cells were digested with 0.25% trypsin. Proteolysis was neutralized with 10% FBS, and the lysates were centrifuged at 3000 rpm for 5 min, washed once with PBS, and stained using the Annexin V-FITC and a propidium iodide (PI) solution (Beyotime Biotechnology, Shanghai, China) for 15 min at room temperature away from light. The percentage of apoptotic cells for each sample was subsequently evaluated by a BD FACSCalibur flow cytometer (BD Biosciences, Franklin Lakes, NJ, USA).

### 4.15. Statistical Analysis

Data analysis was performed by GraphPad Prism (version 8.1 for Windows, GraphPad) and R version 4.0.3 (R Development Team). Data are expressed as the means ± standard errors of the mean (SEMs). Violin plots were generated using the ggplot 2 package (version 2.2.1). *p*-values were calculated using Students *t*-test or analysis of variance (ANOVA) with Dunnett’s test (for one-way ANOVA) or Tukey’s (for two-way ANOVA) multiple-comparison test, with statistical significance as follows: * *p* < 0.05; ** *p* < 0.01; *** *p* < 0.001, **** *p* < 0.001.

## 5. Conclusions

In conclusion, this study demonstrated that melatonin exerts a protective effect on LPS-induced implantation failure and abnormal pregnancy in mice through the MTNR1B-m6A pathway (Figure 8). By targeting this pathway, melatonin alleviated the abnormal increase of inflammatory-related proteins TLR4, MYD88, RELA, autophagy-related proteins BECN1, ATG5, and ATG7, as well as apoptosis-related proteins BAX, CASP9, and CASP3. These findings shed light on the molecular mechanisms underlying melatonin’s beneficial effects on pregnancy and provide valuable insights into potential therapeutic strategies for managing pregnancy-related complications.

## Figures and Tables

**Figure 1 ijms-25-00733-f001:**
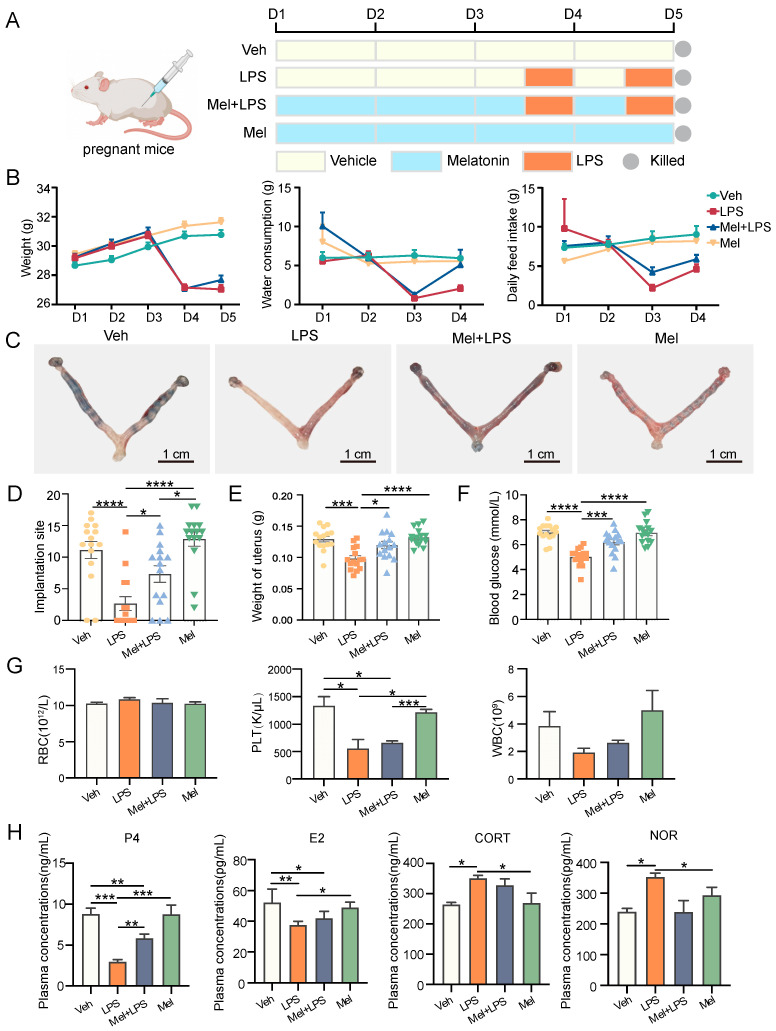
The protective effect of melatonin on mouse embryo implantation. (**A**) Schematic diagram of the animal experimental design. (**B**) Changes in the body weight, water consumption, and daily feed intake of pregnant mice. *n* = 36 independent biological replicates. (**C**) Implantation sites were indicated by injecting Chicago Sky Blue dye. (**D**) The number of implantation sites in mice. *n* = 15 independent biological replicates. (**E**) The weight of uterus in mice. *n* = 15 independent biological replicates. (**F**) The blood glucose values in mice. *n* = 15 independent biological replicates. (**G**) Blood was analyzed for the number of RBC, HB, PLT, WBC. RBC: red blood cell; HB: hemoglobin concentration; PLT: platelet count; white blood cell. *n* = 3 independent biological replicates. (**H**) Serum hormone analysis in mice. *n* = 3 independent biological replicates. P4: progesterone; E2: Estradiol-17β; CORT: corticosterone; NOR: noradrenaline. Veh: vehicle treatment group; LPS: LPS treatment group; Mel+LPS: melatonin and LPS co-treatment group; Mel: melatonin treatment group; The data are presented as the mean ± SD. Levels of statistical significance for all data were determined by one-way ANOVA and Tukey’s test (* Indicates significant difference between the two groups; * *p* < 0.05; ** *p* < 0.01; *** *p* < 0.001; **** *p* < 0.0001).

**Figure 2 ijms-25-00733-f002:**
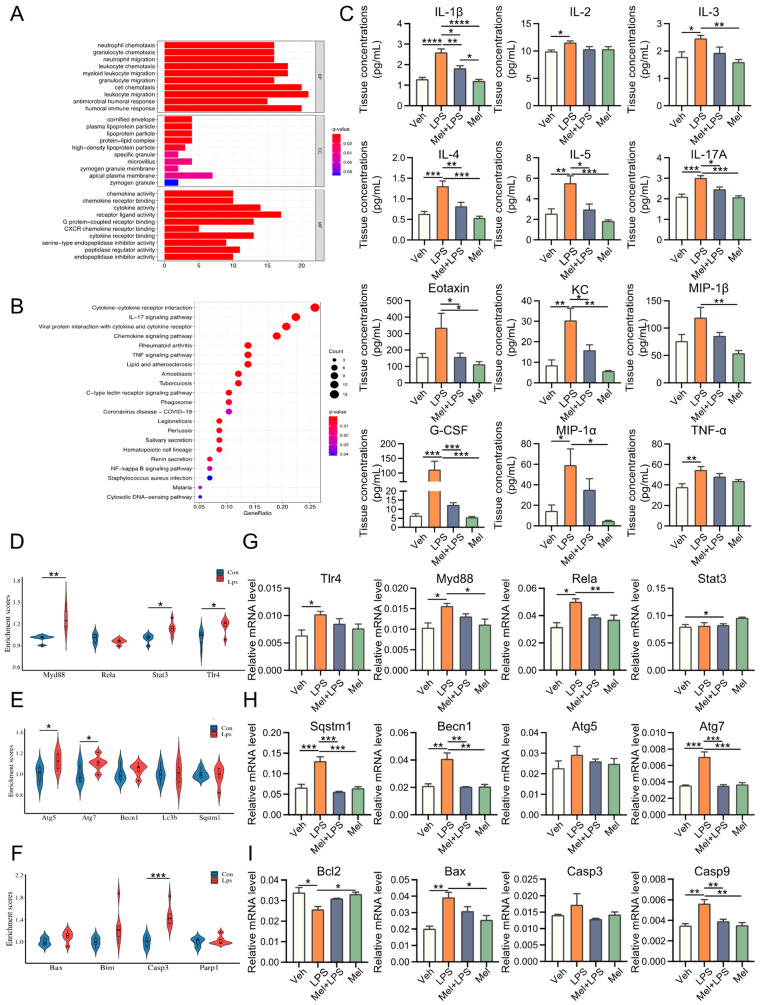
Melatonin alleviates LPS-induced inflammation, autophagy, and apoptosis in uterus. (**A**) Gene Ontology (GO) enrichment analyses of DEGs between Con and Lps. (**B**) Kyoto Encyclopedia of Genes and Genomes (KEGG) pathway enrichment analyses of DEGs between Con and Lps. The top 21 pathways enriched in KEGG. (**C**) Uterus cytokine analysis by Luminex. *n* = 4 independent biological replicates. (**D**) Violin plots show the expression levels inflammation-related genes mRNAs between Con and Lps. (**E**) Violin plots show the expression levels autophagy-related genes mRNAs between Con and Lps. *n* = 3 independent biological replicates. (**F**) Violin plots show the expression levels apoptosis-related genes mRNAs between Con and Lps. *n* = 3 independent biological replicates. (**G**) The mRNA levels of the inflammation-related genes in uterus of mice. *n* = 3 independent biological replicates. (**H**) The mRNA levels of the autophagy-related genes in uterus of mice. *n* = 3 independent biological replicates. (**I**) The mRNA levels of the apoptosis-related genes in uterus of mice. *n* = 3 independent biological replicates. Veh: vehicle treatment group; LPS: LPS treatment group; Mel+LPS: melatonin and LPS co-treatment group; Mel: melatonin treatment group; The data are presented as the mean ± SD. Levels of statistical significance for all data were determined by one-way ANOVA and Tukey’s test (* Indicates significant difference between the two groups; * *p* < 0.05; ** *p* < 0.01; *** *p* < 0.001; **** *p* < 0.0001).

**Figure 3 ijms-25-00733-f003:**
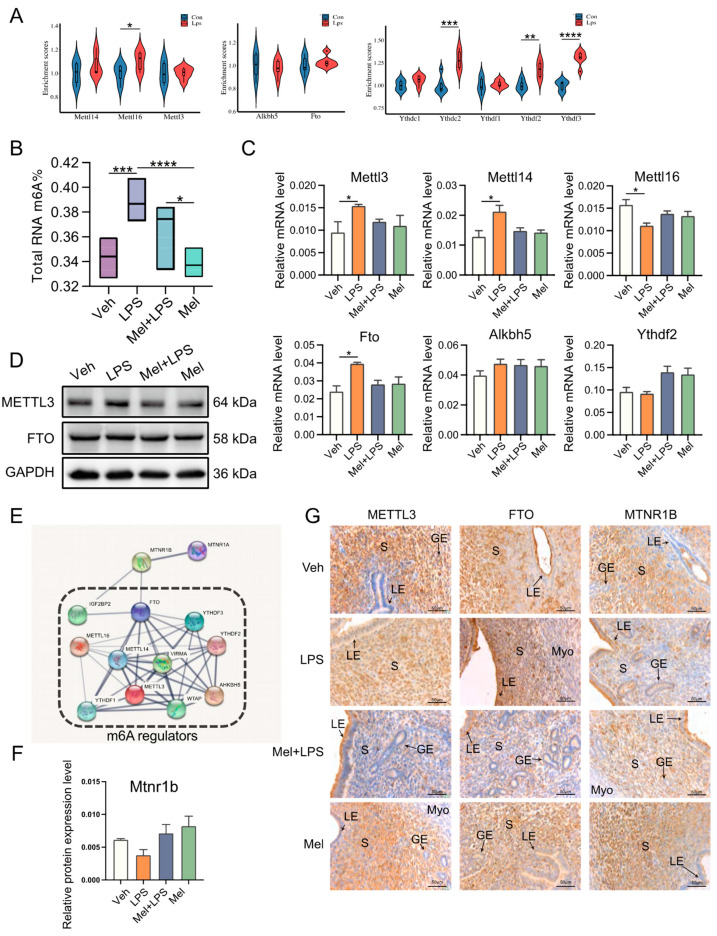
Melatonin alleviates LPS-induced elevated m6A levels in uterus. (**A**) Violin plots show the expression levels of m6A regulaters mRNAs between Con and Lps. *n* = 3 independent biological replicates. (**B**) Global m6A levels in the uterus. *n* = 6 independent biological replicates. (**C**) The mRNA levels of m6A regulaters in the uterus of mice. *n* = 3 independent biological replicates. (**D**) Western blot bands of METTL3 and FTO. *n* = 3 independent biological replicates. (**E**) Protein–protein interaction (PPI) network of the significant genes. Using the STRING online database. (**F**) The mRNA levels of *Mtnr1b* in the uterus of mice. *n* = 3 independent biological replicates. (**G**) Immunohistochemical (IHC) staining of METTL3 and FTO in the uterus on D5. *n* = 3 independent biological replicates. S: stroma; LE: luminal epithelium; GE: glandular epithelium; Myo: myometrium. Veh: vehicle treatment group; LPS: LPS treatment group; Mel+LPS: melatonin and LPS co-treatment group; Mel: Melatonin treatment group; The data are presented as the mean ± SD. Levels of statistical significance for all data were determined by one-way ANOVA and Tukey’s test (* Indicates significant difference between the two groups; * *p* < 0.05; ** *p* < 0.01; *** *p* < 0.001; **** *p* < 0.0001).

**Figure 4 ijms-25-00733-f004:**
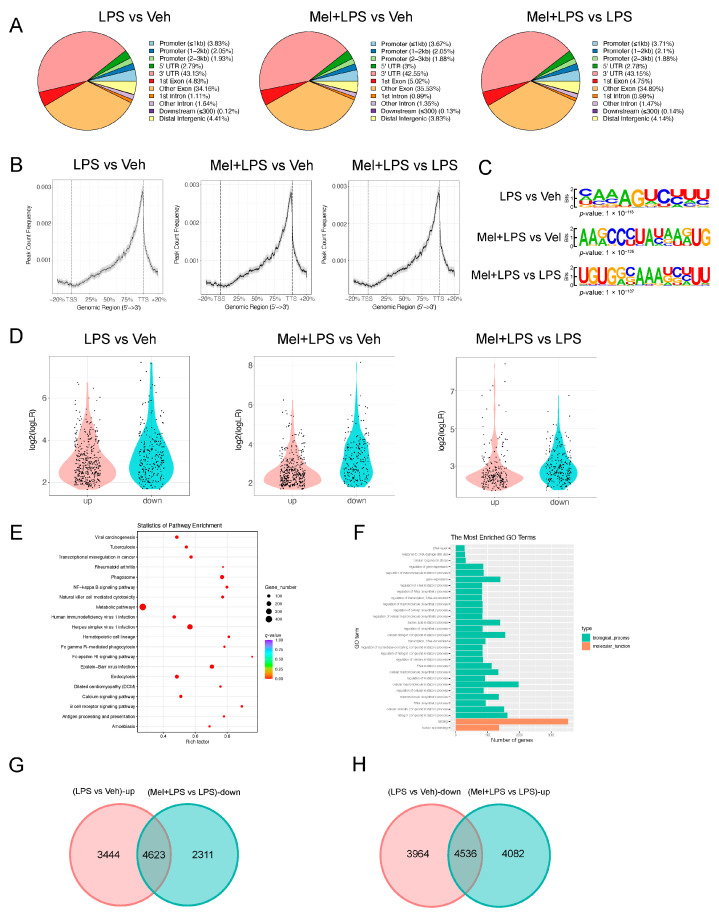
m6A-seq analysis of m6A modification after melatonin and LPS stimulation. (**A**) The annotation of m6A-seq different reads between different treatments. (**B**) Genome browser view of m6A-seq different reads between different treatments. (**C**) The motif of m6A-seq different reads between different treatments. (**D**) Violin plots of the m6A-up and m6A-down genes between different treatments. (**E**) KEGG pathway enrichment of m6A-up genes between LPS group and Veh group. (**F**) GO enrichment of m6A-up genes between LPS group and Veh group. (**G**) Venn diagram of m6A-up genes in the LPS group and m6A-down genes in the Mel+LPS group. (**H**) Venn diagram of m6A-down genes in the LPS group and m6A-up genes in the Mel+LPS group. *n* = 3 independent biological replicates. Veh: vehicle treatment group; LPS: LPS treatment group; Mel+LPS: melatonin and LPS co-treatment group.

**Figure 5 ijms-25-00733-f005:**
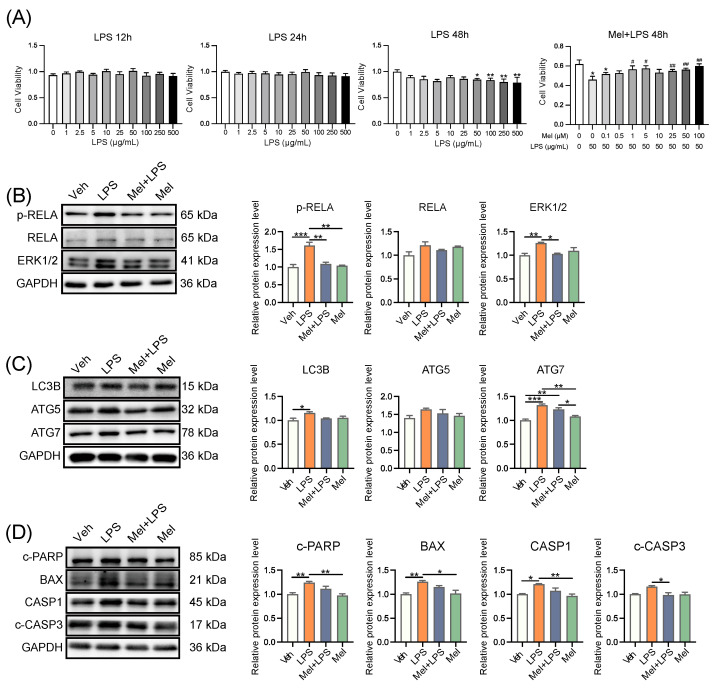
Melatonin alleviates LPS-induced inflammation, autophagy, and apoptosis in human endometrial stromal cells. (**A**) Cell proliferation assay by CCK8 method. *n* = 6 independent biological replicates. LPS: added LPS only; Mel+LPS: added melatonin and LPS; * Indicates a significant difference compared with the LPS 0 μg/mL group; * *p* < 0.05; ** *p* < 0.01; ^#^ Indicates a significant difference compared with the LPS 50 μg/mL group; ^#^
*p* < 0.05; ^##^
*p* < 0.01. (**B**) Western blot bands of inflammation-related proteins in human endometrial stromal cells. *n* = 3 independent biological replicates. (**C**) Western blot bands of autophagy related proteins in human endometrial stromal cells. *n* = 3 independent biological replicates. (**D**) Western blot bands of apoptosis related proteins in human endometrial stromal cells. *n* = 3 independent biological replicates. Veh: vehicle treatment group; LPS: LPS treatment group; Mel+LPS: melatonin and LPS co-treatment group; Mel: melatonin treatment group; The data are presented as the mean ± SD. Levels of statistical significance for all data were determined by one-way ANOVA and Tukey’s test (* Indicates significant difference between the two groups; * *p* < 0.05; ** *p* < 0.01; *** *p* < 0.001).

**Figure 6 ijms-25-00733-f006:**
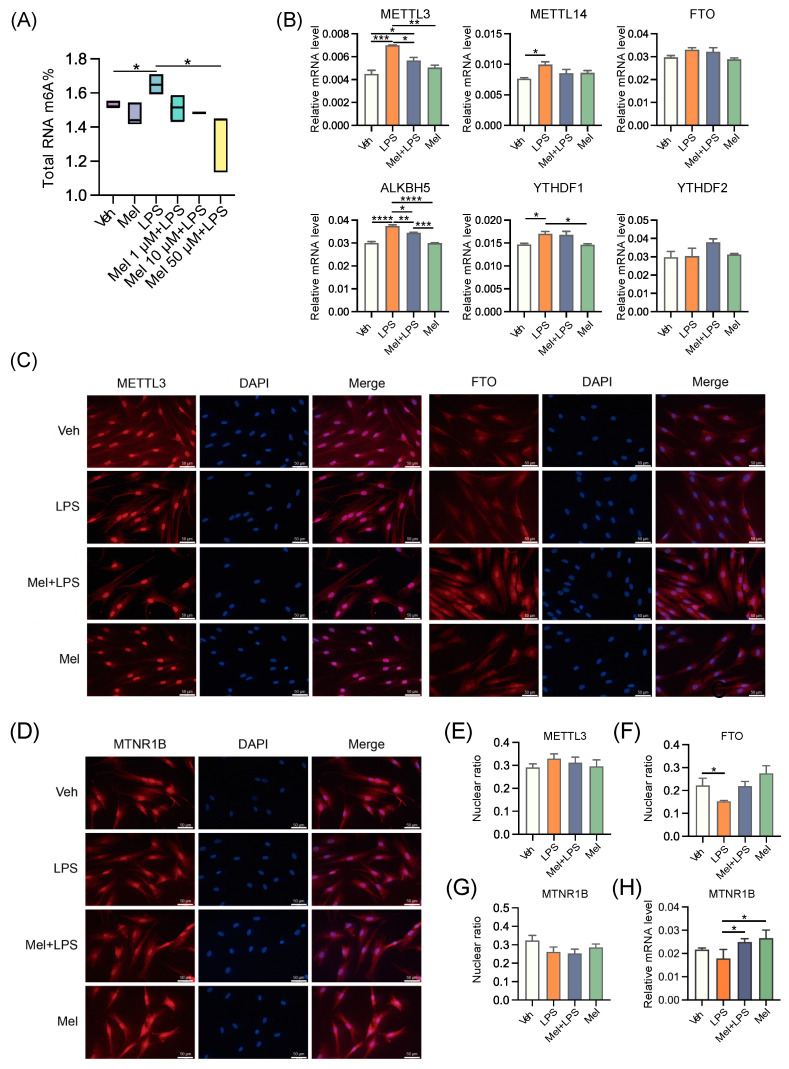
Melatonin alleviates LPS-induced elevated m6A levels in human endometrial stromal cells. (**A**) Global m6A levels of human endometrial stromal cells treated with LPS and different concentrations of melatonin. *n* = 3 independent biological replicates. (**B**) The mRNA levels of the m6A regulators in uterus of human endometrial stromal cells treated with LPS and melatonin. *n* = 3 independent biological replicates. (**C**) Immunofluorescence of m6A-related proteins in HESCs. (**D**) Immunofluorescence of MTNR1B in HESCs. (**E**) The ratio of the fluorescence intensity of METTL3 in the nucleus to the fluorescence intensity of METTL3 in the whole cell. *n* = 6 independent biological replicates. (**F**) The ratio of the fluorescence intensity of FTO in the nucleus to the fluorescence intensity of FTO in the whole cell. *n* = 6 independent biological replicates. (**G**) The ratio of the fluorescence intensity of MTNR1B in the nucleus to the fluorescence intensity of MTNR1B in the whole cell. *n* = 6 independent biological replicates. (**H**) The mRNA levels of *MTNR1B* in the uterus of mice. The experiments were performed in triplicate. The data are presented as the mean ± SD. Levels of statistical significance for all data were determined by one-way ANOVA and Tukey’s test (* Indicates significant difference between the two groups; * *p* < 0.05; ** *p* < 0.01; *** *p* < 0.001; **** *p* < 0.0001).

**Figure 7 ijms-25-00733-f007:**
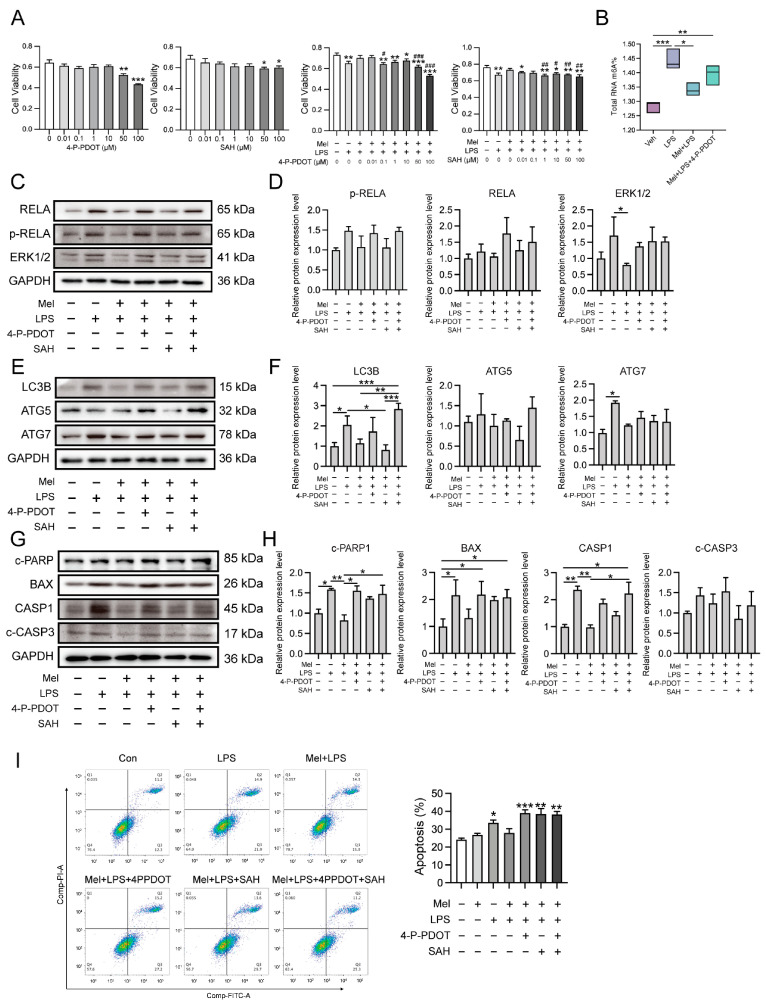
Melatonin plays a protective role through MTNR1B. (**A**) Cell proliferation assay by CCK8 method. *n* = 6 independent biological replicates. 4-P-PDOT: 4-phenyl-2-propionamidotetralin; SAH: S-adenosylhomocysteine; * Indicates a significant difference compared with the 4-P-PDOT/SAH 0 μM group; * *p* < 0.05; ** *p* < 0.01; *** *p* < 0.001; ^#^ Indicates a significant difference compared with the LPS added group alone Difference; ^#^
*p* < 0.05; ^##^
*p* < 0.01; ^###^
*p* < 0.001. (**B**) Global m6A levels of human endometrial stromal cells treated with 4PPDOT. (**C**) Western blot bands of inflammation-related proteins in human endometrial stromal cells. (**D**) Immunoblot analysis of *p*-RELA, RELA, ERK1/2. (**E**) Western blot bands of autophagy related proteins in human endometrial stromal cells. (**F**) Immunoblot analysis of LC3B, ATG5, ATG7. (**G**) Western blot bands of apoptosis related proteins in human endometrial stromal cells. (**H**) Immunoblot analysis of c-PARP, BAX, CASP1, c-CASP3. (**I**) Flow cytometry was used to detect apoptosis after adding the 4PPDOT and melatonin, and apoptosis cells were measured. *n* = 3 independent biological replicates. Veh: vehicle treatment group; LPS: LPS treatment group; Mel+LPS: melatonin and LPS co-treatment group; Mel: melatonin treatment group; The data are presented as the mean ± SD. Levels of statistical significance for all data were determined by one-way ANOVA and Tukey’s test (* Indicates significant difference between the two groups; * *p* < 0.05; ** *p* < 0.01; *** *p* < 0.001).

**Figure 8 ijms-25-00733-f008:**
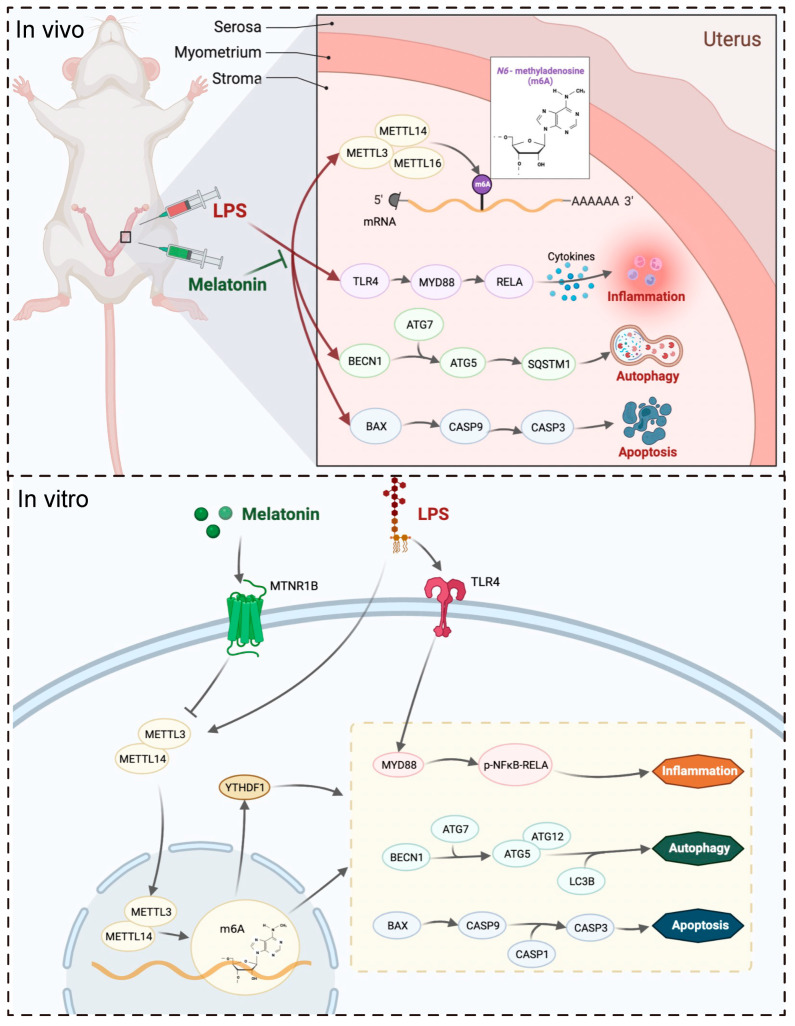
The mechanism by which melatonin protects pregnancy.

## Data Availability

The m6A-seq raw sequence data has been uploaded to Gene Expression Omnibus GSE216994 (https://www.ncbi.nlm.nih.gov/geo/query/acc.cgi?acc=GSE216994) (accessed on 15 March 2023).

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
