# Peer review of "Melatonin Alleviates Lipopolysaccharide-Induced Abnormal Pregnancy through MTNR1B Regulation of m6A"

_ijms, 2024, doi:10.3390/ijms25020733_

Round 1
Reviewer 1 Report
Comments and Suggestions for Authors
ijms-2759945-peer-review-v1
This is an interesting, complex work combining different experimental approaches and results supported with appropriate illustrative material. However, maybe authors can consider to move some of the visual material as supplementary figures, and keep only essential for the following the work figures.
Ln39: Please, add "." at the end of the sentence. (.... function [6].)
Ln40: Please, correct to: rhythms [7]
Maybe the size of all figures can be increased to full page size figure.
Ln182: In this and similar occasions, please, use italics for in vitro, in vivo, etc. other Latin expressions.
Maybe will be appropriate if authors can provide the number of the ethical committee approve for conducting this research.
For the cited supplier of material and equipment, use address of headquarters of the company and not distributors. Please, check all supplied information, and pay attention to the abbreviation of the states. WU is not corresponding to any state in USA, maybe is WV, as West Virginia? However, this is not headquarter address of Sigma-Aldrich. Please, check and correct for entire manuscript.
For all applied equipment’s needs to be provide address of the supplier. Please, see Ln 430 as example and Yoyue blood glucose meter equipment.
Ln450: Add city for Bio-Rad company.
Ln462: Reference need to be cited by number. Please, correct.
Ln466-467: add city as part of address for Invitrogen and Illumina.
Most of the methods are presented in very telegraphic way, Please, provide more details and when applicable reference/s, for the applied experimental procedures
References need to be formatted according to the instructions form the journal.
Reviewer 2 Report
Comments and Suggestions for Authors
The manuscript entitled Melatonin alleviates LPS-induced abnormal pregnancy through MTNR1B regulation of m6A, discussed mechanisms behind how melatonin is effective in the embryo implantation process. authors conducted a wide range of experiments and the results are presented. I have just minor concerns including
1. what is the point of comparing those groups in Venn diagrams (Fig 4 G and H). are they comparable enough?
2. how these results can be translated to the human side needs to be discussed.
3. at least on my computer I had difficulties reading the gene names in the heatmap presented in the supplementary data. worth double checking.
Comments on the Quality of English LanguageEnglish is fine.
